# Role of Surgery in Metastatic Melanoma and Review of Melanoma Molecular Characteristics

**DOI:** 10.3390/cells13060465

**Published:** 2024-03-07

**Authors:** Kulkaew Sukniam, Harsheen K. Manaise, Kyle Popp, Reed Popp, Emmanuel Gabriel

**Affiliations:** 1Department of General Surgery, Duke University Medical Center, Durham, NC 27707, USA; 2Department of Medicine, Government Medical College and Hospital, Chandigarh 160047, India; 3Department of Medicine, Florida State University, Tallahassee, FL 32306, USA; 4College of Medicine, University of Florida, Gainesville, FL 32611, USA; 5Department of General Surgery, Division of Surgical Oncology, Mayo Clinic Florida, Jacksonville, FL 32224, USA

**Keywords:** melanoma, mutation, surgery, metastatic melanoma

## Abstract

We aimed to review the molecular characteristics of metastatic melanoma and the role of surgery in metastasectomy for metastatic melanoma. We performed a systematic literature search on PubMed to identify relevant studies focusing on several mutations, including NRAS, BRAF, NF1, MITF, PTEN, TP53, CDKN2A, TERT, TMB, EGFR, and c-KIT. This was performed in the context of metastatic melanoma and the role of metastasectomy in the metastatic melanoma population. A comprehensive review of these molecular characteristics is presented with a focus on their prognosis and role in surgical metastasectomy.

## 1. Introduction

Melanoma, a type of skin cancer originating from melanocytes, exhibits a dual nature. In its early stages, melanoma is highly treatable and offers a favorable prognosis. However, once metastases develop, the situation becomes more dire, leading to a substantial decline in survival rates. For localized melanoma, the 5-year survival rate is as high as 98%. However, this decreases to 16% for metastatic melanoma [1]. The site of distant metastasis significantly affects prognosis, with visceral melanoma metastases being associated with the poorest outcomes.

In 2023, an estimated 97,610 individuals will be diagnosed with cutaneous melanoma in the United States, and about 7990 will die of metastatic melanoma [2]. In the past, in patients who were diagnosed with melanoma, a complete surgical excision was known to be the gold standard in stage III or IV metastatic melanoma disease [3]. The NCCN Clinical Practice Guidelines in Oncology (NCCN Guidelines) for melanoma cutaneous primary treatment includes wide excision (WE) of primary melanoma with sentinel lymph node biopsy (SLNB). In patients with stage III–IV melanoma, systemic immunotherapy and sometimes targeted surgery are also recommended to improve remission and survival [4]. With advances in medical and surgical oncology, therapeutic lymph node dissection in patients with regional metastatic melanoma has become part of multidisciplinary management. With metastasectomies, it is important to ensure that the tumor is completely excised and that quality of life will not be significantly compromised after the surgery.

As mortality rates in metastatic melanoma remain high, new clinical guidelines are leaning toward comprehensive treatment for advanced metastatic melanoma, and a better understanding of the molecular pathogenesis of malignant melanoma is needed to improve the care of patients with metastatic melanoma.

To this end, we conducted a retrospective review of molecular subtypes of melanoma and the role of metastasectomy among these subtypes. The search period was from 2013 to 2023, with a restriction to English language publications. The search terms “melanoma”, “mutation”, “surgery”, and “metastatic melanoma” and specific gene names (TMB, BRAF, NRAS, EGFR, c-KIT, MITF, etc.) were used in combination with the Boolean operators AND or OR. Four independent investigators (KS, HKM, KP, and EG) performed the literature search and screened the retrieved articles for relevance. We reviewed papers from the years 2013 to 2023, amounting to 81 papers. The initial screening was based on titles and abstracts, followed by a full-text review of potentially relevant articles. The reference lists of the included articles were also reviewed to identify additional relevant literature. The inclusion criteria for the studies selected were as follows: (i) studies examining the selected mutations in metastatic melanoma; (ii) studies containing discoveries on the clinical implications, therapeutic outcomes, or predictive significance of these mutations; (iii) studies presenting the role of surgery in metastatic melanoma; and (iv) studies published in peer-reviewed journals.

## 2. Melanoma Mutations of Importance (Summarized in Table 1)

### 2.1. NRAS

The hyperactivation of the mitogen-activated protein kinase (MAPK) cell signaling pathway can result from oncogenic NRAS (neuroblastoma ras viral oncogene homolog) gene mutations [5,6]. Nodular melanoma and melanomas resulting from persistent UV exposure on the skin frequently have NRAS mutations [7,8,9]. Recent research has demonstrated a significant correlation between NRAS mutations and shorter disease-free survival compared with melanoma patients without these mutations [10]. Additionally, the presence of NRAS mutations is associated with reduced median relapse-free survival and overall survival following the surgical removal of lung metastasis in comparison with mutations in the BRAF, CKIT, and EGFR genes. Among these mutations, only NRAS has been identified as a predictive factor for shorter survival [11]. NRAS mutant tumors are characterized as more aggressive, especially in earlier stages of the disease [9]. Additionally, NRAS mutations have been found to be associated with a shorter time to locoregional nodal relapse and a higher frequency of nodal relapse [9]. Interestingly, NRAS mutant tumors are associated with higher Breslow thickness and Clark levels of invasion, as well as an affinity for tumors on the extremities [5]. A significantly higher mean age at diagnosis has also been found with NRAS mutant tumors [6].

**Table 1 cells-13-00465-t001:** Important mutations in melanoma.

NRAS	NRAS mutation correlates with shorter disease-free survival and reduced overall survival following surgical removal of lung metastasis. NRAS mutant tumors tend to be more aggressive; have higher Breslow thickness and Clark levels of invasion and occur at higher ages.
BRAF	BRAF mutations are prevalent in melanomas, primarily in trunk locations, and targeted therapies have shown clinical responses.
NF1	NF1 inactivation leads to MAPK activation, and NF1 mutations are observed in older patients, desmoplastic melanoma, and males. NF1-mutated melanomas exhibit increased sensitivity to MEK inhibitors.
MITF	Direct inhibition of MITF is challenging, but compounds disrupting MITF dimer formation show promise.
PTEN	PTEN expression correlates with tumor thickness and impacts patient survival rates.
TP53	Elevated TP53 expression is linked to lymph node metastases and reduced survival in melanoma patients.
CDKN2A	CDKN2A status affects response to CDK4/6 inhibitors and immunotherapy.
TERT	TERT promoter mutations are common in melanoma and are associated with aggressive tumors and worse outcomes.
TMB	TMB is a strong predictive factor for relapse-free survival and response to immune checkpoint inhibitors in metastatic melanoma.
EGFR	EGFR mutations activate signaling pathways associated with resistance to BRAF inhibitors and melanoma progression.
C-KIT	KIT mutations are rare in melanoma but can be targeted with therapies like imatinib.

### 2.2. BRAF

BRAF (v-raf murine sarcoma viral oncogene homolog B1), encoded by the BRAF gene on chromosome 7, is a serine/threonine protein kinase and is activated by the Ras-GTP protein. BRAF activates the MAP kinase/ERK signaling pathway. The BRAFV600E mutation leads to the constitutive activation of the kinase and suppresses negative feedback, resulting in the sustained activation of the downstream MEK/ERK pathway. This dysregulation promotes increased cell proliferation, tumor cell invasion, and metastasis [12]. Approximately 50% of melanomas exhibit BRAFV600 mutations, with the majority (up to 90%) of these mutations involving a substitution of valine by glutamic acid at codon 600, referred to as BRAFV600E. This consists of a T1799A transversion mutation in exon 15 of the BRAF gene, resulting in V600E (Val600Glu) amino acid substitution in the protein [13]. Other mutations, including BRAFV600R and BRAFV600D, are exceedingly rare in melanoma. The occurrence of BRAF mutations varies across different tumor sites, with higher frequencies observed in the trunk (57%), followed by the extremities (46%) and the face or scalp (28%) [8].

Targeted therapies aimed at BRAF V600E-mutated melanoma have shown significant clinical responses, although they are often transient. Importantly, the BRAF and NRAS mutations are mutually exclusive [14]. Early studies suggest that BRAF mutation is an early event in melanoma development, representing a distinct subtype of melanoma with unique characteristics and clinical outcomes. In primary melanoma, BRAF mutation is associated with younger age at diagnosis; localization to the trunk; the absence of chronic sun damage; occurrence as an occult or single primary melanoma; and specific histopathological features such as the presence of mitosis, superficial spreading melanoma, and low Breslow thickness. Recent studies indicate that the V600K mutation is the second most common BRAF mutation in melanoma, representing approximately 20–30% of BRAF mutant melanoma tumors [14].

Melanomas that have mutations other than BRAF V600E, such as RAS mutations or non-V600 BRAF mutations, do not exhibit a positive response to BRAF inhibitors. This is particularly evident in mucosal melanoma, as the occurrence of BRAF mutations is less frequent in this specific subtype [15]. Furthermore, ongoing studies are investigating the combination of immune checkpoint inhibitors and targeted therapies, either administered concurrently or sequentially. While the results of these studies have generally shown promise, no official approvals for such combinations have been obtained thus far [16,17,18].

In a parallel development, small-molecule inhibitors like vemurafenib have emerged as powerful tools in the fight against melanoma. Vemurafenib’s mechanism of action involves the selective inhibition of the mutant BRAF protein, with a particular focus on the V600E mutation that is prevalent in approximately 45% of melanoma cases. This targeted approach has shown promise in clinical practice. A recent phase III trial demonstrated substantial improvements in overall survival among BRAF mutant, advanced melanoma patients who received vemurafenib treatment compared with those treated with dacarbazine [16,17,18,19,20].

These advances in therapeutic options represent a shift in the melanoma treatment landscape. The integration of both immunotherapeutic and targeted therapy approaches is reshaping the way this disease can be combated, offering renewed hope to patients with varying genetic profiles. As research continues to unveil novel therapeutic targets and treatment strategies, the prospects for improved outcomes and prolonged survival for melanoma patients increase.

### 2.3. NF1

The NF1 (neurofibromatosis type 1) gene product is a GTPase (guanosine triphosphatase), which is an activating protein that inhibits RAS function. NF1 inactivation results in the activation of MAPK. NF1 mutations have been observed more so in older patients with a high tumor mutational burden, patients with desmoplastic melanoma, and males. Additionally, NF1 mutations tend to occur in sun-exposed places because UV exposure is the primary cause of genetic mutations and creates a unique pattern in this subgroup [19]. Loss-of-function mutations or deletions in NF1 are associated with reduced responsiveness to BRAF inhibitors in BRAF-mutated melanomas. Studies have shown that NF1-mutated melanomas have increased sensitivity to MEK inhibitors [20].

Given the common occurrence of NF1 mutation in BRAF-WT melanoma, which renders patients unsuitable for BRAF targeting, it is crucial to thoroughly understand NF1-mutated melanoma to guide the development of appropriate treatments for this aggressive subtype. NF1-mutated melanoma is associated with the increased expression of CDC20 and MKI67, which play a crucial role in cell cycle progression, as well as LY6E, which is known to increase the production of proangiogenic agents, including VEGF-A [19].

Clinical trials are not currently explicitly conducted for patients with NF1 mutant melanomas. However, ongoing basket clinical trials are investigating targeted therapy in NF1 mutant solid tumors, including melanomas. These trials include the following:The MATCH screening trial, assessing trametinib for treating NF1 mutant refractory solid cancers (NCT02465060);The MatchMel trial, which involves a group with NF1 mutant refractory tumors receiving trametinib, sorafenib, or everolimus (NCT02645149);The examination of RMC4630 (a potent PTPN11 inhibitor) and cobmitinib in solid tumors with NF1 mutations (NCT03634982) [21].

A method has been proposed to increase sensitivity to trametinib in melanoma cell lines by stabilizing the NF1 protein by lowering its degradation. NF1 may be restored by blocking calpain1 (CAPN1), a calcium-dependent neutral cysteine protease involved in NF1 degradation. This suppression leads to the prevention of RAS activation in melanoma cells. Nevertheless, more clinical trials are required to investigate the safety and effectiveness of this method. Currently, the conventional treatment for melanoma patients with NF1 mutations is the use of immune checkpoint therapies [21].

### 2.4. MITF

MITF (microphthalmia-associated transcription factor) controls the growth of melanocytes and acts as an oncogene for melanoma. MITF is crucial for the development, advancement, and recurrence of melanoma and is viewed as a potential target for treatment. Nevertheless, directly blocking MITF with small molecules has been deemed unfeasible because there is no binding site for drug design [22]. Structural investigations demonstrate that the MITF structure is hyperdynamic because of an out-of-register leucine zipper with a three-residue insertion. The MITF protein is susceptible to mutations that impair dimer formation, particularly in cases of MITF loss-of-function mutations, as found in human Waardenburg syndrome type 2A. These mutations often occur on the dimer interface, affecting the protein’s capacity to form dimers. Dimerization may represent a unique treatment approach to block MITF using small-molecule compounds that can interfere with the MITF dimer. Liu et al. identified a compound TT-012 that binds selectively to dynamic MITF, disrupting its dimer formation and capacity to bind to DNA. TT-012 suppresses the proliferation of high-MITF melanoma cells and hinders tumor growth, albeit with some damage to liver and immune cells in animal models [22].

Research has demonstrated that MITF can enhance the lifespan of melanoma cells by utilizing various anti-apoptotic pathways. Currently, there are no direct inhibitors of MITF available. However, various investigations have identified possible treatment candidates that target the upstream regulators and downstream effectors of MITF, including the following:HDAC inhibitors decrease MITF expression and inhibit tumor growth in the human cutaneous melanoma xenograft model;The HDAC trichostatin A (TSA) increases miR-137 expression, reducing MITF expression in uveal melanoma cells;A small molecule inhibitor of p300/CBP significantly reduces human cutaneous melanoma cells in vitro;Both the HDAC inhibitor ACY-1215 and a small-molecule inhibitor of the MITF pathway (ML329) decrease the proliferation of the metastatic uveal melanoma (UM) cell line in vitro [23].

### 2.5. PTEN

PTEN (phosphatase and tensin homolog) is an established tumor suppressor gene. The absence of PTEN surveillance results in elevated phosphorylation and the stimulation of the AKR mouse-transforming (AKT) serine/threonine kinase, commonly called protein kinase B (PKB). The PI3K/AKT pathway is significantly activated in some melanomas, leading to increased malignant tumor development and the suppression of the apoptotic pathway in melanoma. PTEN loss or downregulation leads to immune evasion and escape by hindering immunogenic cell death, which is crucial for activating tumor-specific cytotoxic T lymphocytes (CTL). This contributes to the resistance to T-cell-mediated immunotherapies, like immune checkpoint inhibitors, in these patients.

Sun et al. showed that PTEN expression was markedly reduced in melanoma samples with a vertical tumor thickness exceeding 2.0 mm (T3 and T4) compared with those with a vertical tumor thickness below 2.0 mm (T1 and T2). This finding may indicate that PTEN protein expression could serve as a biomarker associated with melanoma Breslow thickness as evaluated by tumor depth, which significantly impacts patient outcomes and survival rates [24].

### 2.6. TP53

Mutations in one TP53 (tumor protein 53) allele promote skin cancer growth by dominantly inhibiting the expression of UVB-induced lincRNA-p21, leading to the avoidance of UVB-induced cell death. A high prevalence of mutant p53 has been reported in metastatic melanoma compared with primary melanoma tumors [25]. Aggregate data indicate no significant variation in TP53 expression between persons exposed to sunlight and those who are not [25]. Patients exhibiting elevated TP53 expression show a higher probability of having lymph node metastases in one to three axillary lymph nodes. Multiple investigations have shown that TP53 mutations in cutaneous melanoma are associated with increased aggressiveness and unfavorable outcomes. TP53 mutations are more common in thick and ulcerated melanomas, which are associated with poorer prognosis compared with thin and non-ulcerated melanomas [26]. TP53 mutations are also associated with reduced survival and an increased likelihood of metastasis in individuals with cutaneous melanoma.

### 2.7. CDKN2A

The CDKN2A (cyclin-dependent kinase inhibitor 2A) gene is frequently deactivated as a tumor suppressor gene in melanoma. In 1994, it was found that inherited genetic mutations could deactivate CDKN2A. This finding was a significant advancement in melanoma genetics, identifying CDKN2A as a primary predisposing gene responsible for 10–40% of hereditary melanoma cases [27]. Disabling CDKN2A enhances the susceptibility to CDK4/6 inhibitors in certain types of cancer cells, such as melanoma.

Downregulating CDKN2A in BRAF-inhibitor-resistant melanoma cells improves the effectiveness of palbociclib. Some experimental studies on different types of tumors have shown that deleting CDKN2A does not result in significant sensitivity to CDK4/6 inhibitors. Resistance to treatment may be due to heightened cyclin E1–CDK2 and PI3K–mTOR signaling in tumor cells lacking CDKN2A, which suggests that other changes can reduce the effectiveness of CDK4/6 inhibitors [28].

Two clinical trials (NCT02645149 and NCT02159066) are ongoing where standard-of-care melanoma therapy is provided first, followed by a treatment based on the genetic profile following progression. This involves the simultaneous use of BRAF/MEK and CDK4/6 inhibitors for patients with CDKN2A deletions. These trials may support the clinical application of CDK4/6 inhibitors in treating melanoma patients with CDKN2A deletion.

In addition to targeted therapies, the absence of CDKN2A may impact the effectiveness of immunotherapy in melanoma. Developing immune checkpoint inhibitors (ICIs) that enhance T-cell-mediated anti-tumor immune responses has transformed the management of metastatic melanoma. Several studies have evaluated whether the loss of CDKN2A can serve as a biomarker, though results have yielded conflicting outcomes [29]. Several investigations have found no significant correlation between CDKN2A status and the outcome of immune checkpoint inhibitor treatment in melanoma. One study showed a tendency toward better melanoma control in patients with CDKN2A mutations treated with ICI [29].

Furthermore, individuals with hereditary melanoma caused by germline CDKN2A mutations exhibit a more effective response to immunotherapy. Loss of CDKN2A may result in the accumulation of more mutations in melanomas, leading to the production of additional neoantigens and subsequently enhancing immune responses. CDKN2A could serve as a biomarker for ICI sensitivity, endorsing the utilization of ICIs to enhance anti-tumor immune responses in patients with inactive CDKN2A [29].

### 2.8. TERT

Mutations in the TERT (telomerase reverse transcriptase) promoter are likely the most common mutations in melanoma and have been linked to more aggressive melanomas and worse outcomes, indicating that these changes represent an adverse prognostic factor. TERT promoter mutations are linked to BRAF mutations, and both genetic abnormalities are connected to a worse illness prognosis. Delyon et al. discovered that, in a group of patients with BRAF mutations, most melanomas (85%) had a mutation in the TERT promoter. The most common mutations were −124C > T (50%), followed by −146C > T (28%) and −138/−139CC > TT (7%) [30].

## 3. Less Common Melanoma Mutations

### 3.1. TMB

Multiple studies have shown TMB to be a strong predictive factor of relapse-free survival in metastatic melanoma. Patients with a high tumor mutational burden (TMB) have seen favorable outcomes with immune checkpoint inhibitor (ICI) treatments for multiple metastatic cancers [31]. Patients with elevated TMB have had success with ICI therapies [32,33,34], but for patients with more a targeted treatment, the results are better at lower TMB values. A study utilizing adjuvant dabrafenib and trametinib therapy found patients with a low TMB had prolonged relapse-free survival when compared with patients with high-TMB tumors [35].

### 3.2. EGFR

EGFR (epidermal growth factor receptor) expression in melanomas has been found to impact metastasis to regional lymph nodes, thus indicating its association with tumor aggressiveness [36]. The gene copy amplification of EGFR has also been correlated with melanoma progression and metastasis, with a prevalence of 55% in thicker cancers [37]. Studies have demonstrated that the overexpression of EGFR in melanoma cells promotes their motility and proteolytic activity, ultimately modulating tumor invasiveness [38]. Furthermore, EGFR expression in melanoma has been positively correlated with a poor prognosis for survival [39]. The role of EGFR in melanoma progression has been confirmed using Xmrk, a mutant EGFR derived from Xiphophorus fish, which induces malignant melanoma in medaka fish. This finding holds clinical significance given the high similarity between fish and human melanoma at the ultrastructural and histopathological levels [40].

EGFR with constitutively active mutations leads to the continuous activation of the STAT3 pathway [41]. In BRAF mutant melanoma cells, the STAT3 pathway is directly activated by PI3K-AKT [42]. The EGFR-SFK-STAT3 signaling cascade plays a crucial role in the development of resistance to BRAF inhibitors in melanoma cells [43]. EGFR-STAT3 signaling has been demonstrated to induce melanoma invasiveness [44]. EGFR-induced RAS/AKT/MEKK3 and NF-κB signaling inhibit apoptosis in melanoma cells. RAS phosphorylation upregulates EGFR through an autocrine loop, leading to neoplastic lesion development in transgenic mice [45].

The Xmrk protein, similar to EGFR, activates the PI3K-AKT and RAS-RAF-MEK-MAPK signaling pathways, contributing to tumor progression [46]. The STAT5 pathway is strongly activated during melanoma, inhibiting apoptosis and enhancing proliferation [40]. Small-molecule inhibitors of the epidermal growth factor receptor (EGFR) act by targeting the intracellular tyrosine kinase domain, competing with adenosine triphosphate (ATP) and disrupting downstream signaling pathways mediated by this receptor. First-generation inhibitors, such as gefitinib and erlotinib, are reversible and well tolerated by patients. In contrast, second-generation inhibitors like dacomitinib and afatinib form irreversible covalent bonds within the ATP pocket of the tyrosine kinase domain, thereby enhancing treatment effectiveness. However, their administration is limited because of potential side effects. Third-generation inhibitors, including osimertinib, were designed to target the mutated form of the receptor (T790M) associated with resistance. While EGFR expression in melanoma may exhibit conflicting data, studies have indicated its role in disease progression. For instance, EGFR gene amplification has been linked to a worse prognosis, suggesting its importance in melanoma [24].

Cetuximab, an anti-EGFR monoclonal antibody, has demonstrated its ability to suppress metastasis formation in SCID mice [24]. Matrigel invasion assays have further revealed that cetuximab treatment can significantly reduce the invasive capacity of melanoma cells, such as BLM melanoma cells [24]. This reduction in invasiveness demonstrates the potential of cetuximab as a therapeutic adjuvant for melanoma patients. However, it is important to note that identifying specific markers that predict optimal responses to anti-EGFR therapy remains a significant challenge. Melanoma cells with the constitutive activation of downstream signaling pathways due to oncogene mutations (e.g., K-Ras or B-Raf) or the loss of tumor suppressor genes (e.g., PTEN) may not respond as effectively to cetuximab treatments [24]. The presence of mutations in N-Ras, B-Raf, and PTEN in a subset of melanoma patients suggests that single-agent cetuximab treatments may not be universally effective, highlighting the need for a personalized approach to therapy [24]. Moreover, experiments with EGFR-expressing BLM melanoma cells have shown that cetuximab can reduce invasiveness without compromising cell viability or growth.

### 3.3. c-KIT

c-KIT (CD117 or cluster of differentiation 117) is a glycosylated transmembrane protein represented by an N-terminal extracellular region with five immunoglobulin-like domains, a transmembrane region, and an intracellular tyrosine kinase domain at the C-terminus. Once c-KIT becomes activated by its cytokine ligand, the c-KIT proto-oncogene then plays a significant role in cell proliferation; differentiation; migration and apoptosis (especially in hematopoiesis); stem cell maintenance; gametogenesis; melanogenesis; mast cell development; migration; and function [47]. In melanoma, the c-KIT mutation is rare, occurring in only approximately 3% of melanomas overall [48]. It is often detected in acral melanomas, mucosal melanomas, and chronically sun-damaged skin areas.

A study by Carvajal and Meng showed the clinical benefits of c-KIT targeted therapy in select patients. The guideline from the National Comprehensive Cancer Network recommends the analysis of c-KIT alterations from biopsy samples during the clinical work-up in addition to testing for BRAF mutations. If activating mutations of c-KIT are confirmed in tumors, the c-KIT inhibitor imatinib is currently recommended as one of the second-line systemic therapies after first-line immunotherapy [49]. However, if metastatic tissue is not available, other metastatic sites (e.g., lymph nodes) or the primary tumor may be used because there is minor discordance in BRAF and NRAS mutation profiles between primary and metastatic sites [50,51]. In addition, ipilimumab has been shown to be capable of inducing long-lasting responses and was approved by the FDA for patients with advanced melanoma with c-KIT mutations in first- and second-line treatments. A study by Queirolo and Spagnolo showed that ipilimumab could become a standard treatment for metastatic melanoma, both as a single agent and in combination [52].

## 4. The Role of Metastectomy in Melanoma

At the time of diagnosis, approximately 4% of patients with melanoma have distant metastasis [53]. Several variables, including tumor pathology, mutations (as summarized above), and biomarkers, have been shown to affect the ability of melanoma cells to metastasize, which, in turn, affects patient outcomes [54]. Breslow thickness and ulcerations of the tumor have a substantial effect on the prognosis of stage III melanoma patients with lymphatic metastasis [54], whereas, for stage IV patients, sites of metastasis and serum lactate dehydrogenase (LDH) levels have been established as additional prognostic predictors. Additionally, S100B, a damage-associated molecular pattern protein released by melanoma cells, has been shown to enhance their spread [54].

Bucheit et al. reported that the time between the initial diagnosis of melanoma and the diagnosis of stage IV disease was significantly shorter for patients with the BRAF V600K mutation than for those with the V600E mutation, indicating that BRAF V600K mutations have greater potential for metastatic disease [55,56,57]. However, Chen et al. found that BRAF V600E mutations are more common than BRAF V600K mutations when comparing lymph node samples of patients with samples of their other tissues [58]. These studies may impact individualized treatment plans with respect to the available targeted agents.

NRAS mutation status is associated with metastases to visceral organs other than the lungs, particularly the liver and brain [57]. In addition, NRAS mutations are associated with reduced survival after stage IV diagnosis [59]. Furthermore, Doma et al. found that the fraction of BRAF mutant alleles in metastases in the lungs, adrenal glands, intestines, and kidneys was higher (>40%) than in the primary tumor, highlighting that tumor heterogeneity increases as metastases progress [60].

While surgery is no longer the first-line therapy for most patients with distant metastases, prolonged survival with systemic therapy has provided more prospective candidates for surgery at later stages, both for palliative and curative intent [61]. The role of metastasectomy in melanoma treatment varies significantly depending on the site of metastasis and the patient’s unique clinical presentation. The ability to offer metastasectomy may be highly dependent on the mutational profile of the metastases and the availability of targeted systemic agents to decrease the mutational burden such that surgery may be a viable option.

Gastrointestinal metastasis in melanoma is characterized by a variety of symptoms, including pain (in 29–55% of cases), obstruction (in 27% of cases), bleeding (in 27% of cases), the presence of a palpable mass (in 12% of cases), and unexplained weight loss (in 9% of cases) [62]. Diagnosing GI metastasis is often challenging before surgery, mainly because these symptoms lack specificity and can be attributed to many other medical conditions [63]. In response to these distressing symptoms, palliative interventions frequently become necessary to alleviate these issues. Small-bowel metastasis is the most common site of GI melanoma metastasis, partly because of its rich blood supply [63]. Given the relatively low risk of surgical complications and harm to patients, palliative surgery is a reasonable approach, especially when the patient’s quality of life could be substantially improved, even if achieving long-term, disease-free survival remains an unlikely goal [64]. Importantly, symptomatic recurrence, while a possibility, does not negatively impact outcomes, provided that complete resection remains a feasible option [62]. This highlights the importance of a balanced approach considering symptom relief and the patient’s overall well-being when deciding on surgical intervention for gastrointestinal metastasis

Liver metastasis in melanoma has a poor prognosis, often resulting in an average life expectancy of just 2–4 months [62]. However, liver resection, though not a standard procedure in these cases, may prove beneficial when it can completely remove the metastatic lesions in an R0 resection. A noteworthy study involving a substantial cohort of patients with liver metastasis emphasized the advantages of complete resection [65]. Those who underwent the complete metastasectomy experienced a median survival of 28 months in contrast to counterparts who underwent exploration alone, showing a median survival of only 4 months [65]. Furthermore, another study reinforced the importance of surgical resection whenever feasible. This study revealed that surgical resection, when attainable, has the potential to nearly double the survival rate. It is currently the most effective means of significantly improving the prognosis for individuals with metastatic uveal melanoma [66]. These studies highlight that metastasectomy may have an important role in the multidisciplinary management of metastatic melanoma for more aggressive tumors with more difficult-to-treat mutations (e.g., uveal melanomas).

Brain metastasis is a formidable metastatic site of melanoma, often associated with a poor prognosis, where the average survival is just 4–6 months [62]. Surgical interventions for brain metastasis primarily serve the palliative purpose of alleviating distressing symptoms caused by brain masses, including nausea, visual disturbances, and debilitating headaches, thus enhancing the patient’s quality of life. One of the primary indications for surgical intervention is the presence of a large brain lesion causing severe or impending neurological compromise [67]. Surgery is strongly advocated for single brain metastases, especially in cases where the patient’s systemic disease is controlled or controllable [62]. Melanoma brain metastases are less well studied with respect to mutational burdens than some of the other distant metastatic sites.

The lungs are the most prevalent site for distant metastasis in melanoma cases [68]. Surgical interventions to address lung metastases arising from melanoma are a subject of ongoing debate within the medical community, primarily because of the generally less favorable outcomes compared with primary tumors of other origins, likely also driven by more aggressive tumor mutations. The decision to pursue surgery for melanoma metastasis in the lungs is influenced by various factors, including the extent of resection the time elapsed between the initial diagnosis and the appearance of lung metastases (known as the time to pulmonary metastases or TPM); and the number of metastatic lesions. Specifically, patients with a solitary lung lesion and/or a longer TPM and those who undergo radical resection show improved survival rates following surgical treatment [69]. Recent research, exemplified by Leo et al.’s study, highlights these factors’ critical role in determining patient outcomes. Conversely, individuals with multiple lung metastases and a shorter TPM have poorer outcomes, with surgery offering little benefit. These findings underscore the importance of adopting a methodical approach to patient selection, where considerations such as the number of lesions and TPM are carefully weighed when contemplating surgical interventions for pulmonary metastatic melanoma. Such an approach ensures that patients with the most significant potential for benefit are identified and that surgical resources are allocated optimally to improve overall treatment efficacy.

Many studies have demonstrated that the overall survival of patients undergoing combined treatment with surgery and immunotherapy is comparable to that of patients undergoing surgery alone, which is significantly longer than that of patients receiving immunotherapy alone or no immunotherapy [70,71,72,73]. The widespread use of modern therapies has enabled clinicians to identify patients with metastatic lesions resistant to systemic therapy at a specific organ site. Metastasectomy may serve as an adjunct to systemic treatment in these patients. In addition, the increased use of minimally invasive techniques has decreased morbidity and improved patient outcomes associated with surgery [74,75,76]. A single site of metastasis and a long DFI before metastasis are also associated with better outcomes [74,77,78].

Adoptive cell therapy, a tailored therapy involving the direct administration of tumor-reactive lymphocytes to the patient, is another advantage of surgery in patients with metastatic melanoma [77]. This therapy has demonstrated long-lasting, complete responses. Surgically resected tumor deposits are cultured with high-dose IL-2 to proliferate tumor-infiltrating lymphocytes (TILs). After lymphodepletion with aggressive chemotherapy, TILs with inadequate growth and anti-tumor reactivity are selectively expanded and infused into the patient. This process has also demonstrated efficacy in patients with cerebral metastases. However, selecting appropriate targets for TIL harvesting, such as superficial subcutaneous or lymph node metastases, is essential while avoiding brain or bowel metastases because of recuperation and contamination issues [77].

It is important to note that, while many of the studies of metastasectomy for melanoma have demonstrated the benefits of metastasectomy, there is likely implicit bias in the reporting of these data. Our literature search (between 2013 and 2023) essentially yielded all positive studies. However, those studies that showed no benefit of metastasectomy are much less likely to be reported, which is a common occurrence with negative studies in general. However, some studies, even with positive (albeit modest) reports of benefits from metastasectomy, caution about the role of metastasectomy in metastatic melanoma because of this bias [53,78,79,80]. Overall, it is imperative to not only consider this bias for metastasectomy but also recognize that metastasectomy may provide benefits for highly select patients with metastatic disease, particularly for those in whom systemic therapies, often based on tumor mutational targets, may have minimized the tumor burden.

## 5. Future Directions

The current landscape of melanoma research, with its exploration of NRAS, TMB, BRAF, EGFR, c-KIT, and other mutations, offers a promising trajectory for future endeavors in understanding and combatting this disease. Moving forward, a holistic approach that combines insights from these diverse mutations is essential. As personalized medicine gains traction, the task remains to identify the optimal management approach for individual patients based on their specific mutation profiles and whether surgery may play a role depending on the efficacy of mutation-specific systemic targeted agents. The quest for predictive markers that determine treatment responses should also be an important focus, with the goal of tailoring interventions to maximize positive outcomes.

## 6. Conclusions

In conclusion, we reviewed metastatic melanoma’s genetic and biological properties, highlighting the significance of molecular mutations like NRAS, BRAF, NF1, and others in disease progression and treatment responses. Moreover, we have underscored the evolving role of metastasectomy in specific sites, offering palliative relief and, in some cases, curative potential. Combining surgery with immunotherapy and other targeted therapies shows promise, emphasizing the need for a multidisciplinary approach. It is clear that these extensive mutations interact with cellular processes to cause tumorigenesis. Of note, it has been found that metastatic melanoma tumors have more aggressive behavior when compared with primary tumors. This likely implies that additional mutations may occur during the progression of melanoma. We have identified various downstream cellular pathways that play a role in melanoma. Hence, there is potential to identify targets for immunotherapy drugs and biomarkers to monitor treatment response, which is the subject of many ongoing and future investigations.

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
