# Peer review of "Role of Surgery in Metastatic Melanoma and Review of Melanoma Molecular Characteristics"

_cells, 2024, doi:10.3390/cells13060465_

Round 1
Reviewer 1 Report
Comments and Suggestions for Authors
Dear authors
Your proposal has two clearly differentiated parts. One that refers to how surgical excision of metastases affects survival in metastatic melanoma, and a second part regarding the molecular characteristics that affects prognosis. In opinion of this referee none of the proposed objectives have been reached by the authors.
Most of what is discussed in the work are mere truisms, as is the fact that “a metastasectomy is never performed, unless there is a clear benefit in the patient's quality of life”. It is not necessary to read a review to gain knowledge of the topic. Moreover, the information provided is insufficient and the references used should be updated. To cite some examples:
a) Why do you only mention gastric metastases and not others detected in the abdominal cavity? I suggest reviewing: Ionescu et al (2022) Intra-Abdominal Malignant Melanoma: Challenging Aspects of Epidemiology, Clinical and Paraclinical Diagnosis and Optimal Treatment-A Literature Review. Diagnostics (Basel)
b) Regarding the surgical resection of gastric metastases, there is much more recent and relevant literature than Am J Surg. Published on 1991.
According to the introduction: “To this end, we conducted a retrospective review of molecular subtypes of melanoma and the role for metastasectomy among TMB, BRAF, NRAS, EGFR, c-KIT and MITF melanoma subtypes”. Why MITF in not mentioned in the article latter? Why just these subtypes? Why not others like NF1, PTEN, TP53, CDKN2A, and TERT? Moreover, it must be added that authors do not detail the mechanisms involved in the more aggressive behaviour of the reviewed subtypes.
Regarding chemotherapeutic treatment of the different subtypes of melanoma, again the information provided is insufficient, incomplete and no relevant. To make an example, in patients with metastatic lesions bearing BRAF mutations, the authors do not mention the widely known benefits of the combined BRAF and MEK inhibition. Please check: Liu et al. (2017) Efficacy and safety of BRAF inhibition alone versus combined BRAF and MEK inhibition in melanoma: a meta-analysis of randomized controlled trials. Oncotarget.
Previous reviews are more consistent with the information provided by the present work.
Based on my assessment, this review should be rejected.
Author Response
Thank you to each of the reviewers for their time and highly thoughtful comments. We have tried to incorporate their feedback as much as possible within the scope of our review. Whereas Reviewer 1 and 2 were highly favorable, we recognize and acknowledge the points raised by Reviewer 3, but feel that there is still sufficient value to our manuscript. Thank you for your continued time and consideration of our submission.
Reviewer 3
1 - Your proposal has two clearly differentiated parts. One that refers to how surgical excision of metastases affects survival in metastatic melanoma, and a second part regarding the molecular characteristics that affects prognosis. In opinion of this referee none of the proposed objectives have been reached by the authors.
We appreciate the input from this reviewer. Our goal was to provide an overview of the molecular characteristics and metastasectomy. As an invited article, we proposed a “mini-review” and so did not get highly in-depth. In addition, there is limited data on this topic, but we searched the literature as extensively as possible. We appreciate the other references provided by the reviewer, and have included them in the revisions.
2 - Most of what is discussed in the work are mere truisms, as is the fact that “a metastasectomy is never performed, unless there is a clear benefit in the patient's quality of life”. It is not necessary to read a review to gain knowledge of the topic. Moreover, the information provided is insufficient and the references used should be updated. To cite some examples:
- a) Why do you only mention gastric metastases and not others detected in the abdominal cavity? I suggest reviewing: Ionescu et al (2022) Intra-Abdominal Malignant Melanoma: Challenging Aspects of Epidemiology, Clinical and Paraclinical Diagnosis and Optimal Treatment-A Literature Review. Diagnostics (Basel)
- b) Regarding the surgical resection of gastric metastases, there is much more recent and relevant literature than Am J Surg. Published on 1991.
We have added these references as suggested.
3 - According to the introduction: “To this end, we conducted a retrospective review of molecular subtypes of melanoma and the role for metastasectomy among TMB, BRAF, NRAS, EGFR, c-KIT and MITF melanoma subtypes”. Why MITF in not mentioned in the article latter? Why just these subtypes? Why not others like NF1, PTEN, TP53, CDKN2A, and TERT? Moreover, it must be added that authors do not detail the mechanisms involved in the more aggressive behaviour of the reviewed subtypes.
We have added these additional mutations of interest.
4 - Regarding chemotherapeutic treatment of the different subtypes of melanoma, again the information provided is insufficient, incomplete and no relevant. To make an example, in patients with metastatic lesions bearing BRAF mutations, the authors do not mention the widely known benefits of the combined BRAF and MEK inhibition. Please check: Liu et al. (2017) Efficacy and safety of BRAF inhibition alone versus combined BRAF and MEK inhibition in melanoma: a meta-analysis of randomized controlled trials. Oncotarget.
We appreciate this comment. Systemic treatment, including chemotherapy/immunotherapy/targeted therapies, is not really the focus of this review.
5 - Previous reviews are more consistent with the information provided by the present work.
Based on my assessment, this review should be rejected.
We recognize that we may not be able to address all of these comments, but hope that we were able to address some and clarify others. We hope that based on all the reviews as a whole that our mini-review will still be considered.
Reviewer 2 Report
Comments and Suggestions for Authors
The manuscript entitled "Role of Surgery in Metastatic Melanoma and Review of Melanoma Molecular Characteristics" is a review of the molecular characteristics of metastatic melanoma and the role of surgery in metastasectomy for metastatic melanoma". The review is clear and well-organized. There are a few points to clarify.
1 - Please, rewrite the e conclusion section adding some critical points of view (opinions) of the studies reported. Some future perspectives will be welcome.
2 - Please, add information regarding the number of papers and years covered by the review.
3 - Please, I think that mainly figures and/or tables will increase the visibility of the manuscript.
Author Response
Thank you to each of the reviewers for their time and highly thoughtful comments. We have tried to incorporate their feedback as much as possible within the scope of our review. Whereas Reviewer 1 and 2 were highly favorable, we recognize and acknowledge the points raised by Reviewer 3, but feel that there is still sufficient value to our manuscript. Thank you for your continued time and consideration of our submission.
Reviewer 1
1 - Please, rewrite the e conclusion section adding some critical points of view (opinions) of the studies reported. Some future perspectives will be welcome.
We have added to the conclusion section (highlighted), and added some points of interest (new table) as well.
2 - Please, add information regarding the number of papers and years covered by the review.
We have added the necessary information in the introduction section (highlighted).
3 - Please, I think that mainly figures and/or tables will increase the visibility of the manuscript.
We have added 1 table summarizing the different mutations.
Reviewer 3 Report
Comments and Suggestions for Authors
This review is well written and wants to combine molecular information about melanoma and surgical treatment options in metastatic disease. I do very much appreciate this approach in that it highlights that melanoma is not a disease with one common etiology and therefore it is needed to combine tumor biology with clinical treatment decisions.
However, the choice of NRAS, BRAF is very clear, but c-KIT, TMB and EGFR is maybe not the most logical. I would suggest to refer to the 'molecular classification', which is NRAS, BRAF, NF1 and triple-wild type or (my personal preference) the WHO skin tumours which refers to different pathway of which BRAF en NRAS are considered the 'common' pathway, most likely originating from a common naevus, the 'high mutational burden group (so the TMB) of melanoma developing in chronic sundamaged skin, and the acral and mucosal melanoma (more often with KIT, but also with more CNVs than mutations). There is also a Spitz and a 'blue' pathway which are far less likely to metastasize and are more rare. I think that uveal melanoma (related to blue) are maybe interesting to mention (often the cause of the liver metastasis), but it can also be left out when the focus is really on cutaneous melanoma. EGFR is maybe something to only mention briefly or more information should be provided why this is highlighted.
So my main concern is the choice of which molecular tumor biology is descibed is be based on some literature. Now it seems a bit of a random choice.
Author Response
Thank you to each of the reviewers for their time and highly thoughtful comments. We have tried to incorporate their feedback as much as possible within the scope of our review. Whereas Reviewer 1 and 2 were highly favorable, we recognize and acknowledge the points raised by Reviewer 3, but feel that there is still sufficient value to our manuscript. Thank you for your continued time and consideration of our submission.
Reviewer 2
1 - However, the choice of NRAS, BRAF is very clear, but c-KIT, TMB and EGFR is maybe not the most logical. I would suggest to refer to the 'molecular classification', which is NRAS, BRAF, NF1 and triple-wild type or (my personal preference) the WHO skin tumours which refers to different pathway of which BRAF en NRAS are considered the 'common' pathway, most likely originating from a common naevus, the 'high mutational burden group (so the TMB) of melanoma developing in chronic sun damaged skin, and the acral and mucosal melanoma (more often with KIT, but also with more CNVs than mutations). There is also a Spitz and a 'blue' pathway which are far less likely to metastasize and are more rare. I think that uveal melanoma (related to blue) are maybe interesting to mention (often the cause of the liver metastasis), but it can also be left out when the focus is really on cutaneous melanoma. EGFR is maybe something to only mention briefly or more information should be provided why this is highlighted.
We have added new sections for some of the recommended mutations of interest (highlighted). However, we feel that Spitz nevus and blue pathway are beyond the scope of our review.
Round 2
Reviewer 1 Report
Comments and Suggestions for Authors
Dear editors,
In my opinion, the work has improved significantly compared to the first edition, but still, it lacks great relevance. In addition, several details need to be addressed before publication:
1. Review spelling and grammatical errors.
2. Authors excessively emphasize the review on studies demonstrating the positive impact of metastasectomy on the prognosis of patients with metastatic melanoma. To ensure objectivity, it is advisable to incorporate additional examples from the bibliographic references, where no benefit has been observed or even where counterproductive effects have been demonstrated.
Author Response
In my opinion, the work has improved significantly compared to the first edition, but still, it lacks great relevance. In addition, several details need to be addressed before publication:
We appreciate the reviewer for their continued consideration of our manuscript. We respect their perspective, and thank them for the additional minor comments.
1. Review spelling and grammatical errors.
We have gone through the entire manuscript and made dozens of changes to correct spelling and grammar. These are marked with track changes.
2. Authors excessively emphasize the review on studies demonstrating the positive impact of metastasectomy on the prognosis of patients with metastatic melanoma. To ensure objectivity, it is advisable to incorporate additional examples from the bibliographic references, where no benefit has been observed or even where counterproductive effects have been demonstrated.
We agree with this and have added a new last paragraph to the Metastasectomy section (highlighted in yellow). Essentially all of the studies published on this topic during our defined search period (2013-2023) are positive. This is not unexpected given that positive studies tend to be published more so than negative studies. But we did include examples of studies showing modest benefits, which also caution about bias in these studies. Overall, we tried to balance the positivity of these studies as requested.
Reviewer 3 Report
Comments and Suggestions for Authors
No more comments.
Author Response
No more comments.
Thank you, we appreciate your efforts and have submitted a new manuscript addressing Reviewer #1 comments.